# Extended validation of Aeolus winds with wind-profiling radars in Antarctica and Arctic Sweden

Sheila Kirkwood[1], Evgenia Belova[1], Peter Voelger[1*], Sourav Chatterjee[2], Karathazhiyath Satheesan[3]

1 Swedish Institute of Space Physics, Kiruna, SE-98128, Sweden
2 National Centre for Polar and Ocean Research, Ministry of Earth Sciences, Vasco da Gama, Goa, 403804, India
3 Department of Atmospheric Sciences, School of Marine Sciences Cochin University of Science and Technology, Cochin, Kerala, 682 016, India

*Correspondence to: Peter Voelger  (peter.voelger@irf.se)

**Abstract.** Winds from two wind profiling radars, ESRAD in Arctic Sweden and MARA on the coast of Antarctica, are compared with collocated (within 100 km) winds measured by the Doppler lidar onboard the Aeolus satellite for the time period July 2019 - May 2021 (baseline 2B11). Data is considered as a whole, and subdivided into summer/winter and ascending (afternoon) /descending (morning) passes. Mean differences (bias) and random differences are categorised (standard deviation and scaled median absolute deviation) and the effects of different quality criteria applied to the data are assessed, including the introduction of the 'modified Z-score' to eliminate gross errors. This last criterion has a substantial effect on the standard deviation, particularly for Mie winds. Significant bias is found in two cases, for Rayleigh/descending winds at MARA (-1.4 (+0.7) ms$^{-1}$) and for all Mie winds at ESRAD (+1.0 (+0.3) ms$^{-1}$). For the Rayleigh wind bias at MARA, there is no obvious explanation for the bias in the data distribution. The Mie wind error with respect to the wind data measured at ESRAD shows a skewed distribution toward positive values (Aeolus HLOS wind > ESRAD wind). Random differences (scaled median absolute deviation) for all data together are 5.9 / 5.3 ms$^{-1}$ for Rayleigh winds at MARA/ESRAD respectively, and 4.9 / 3.9 ms$^{-1}$ for Mie winds. When the comparison is restricted to Aeolus measurements with mean location within 25 km from the radars, there is no change to the random differences for Rayleigh winds but for Mie winds they are reduced to 3.3/3.6 ms$^{-1}$. These represent an upper bound for Aeolus wind random errors since they are due to a combination of spatial differences, and random errors in both radar winds and Aeolus winds. The random errors in radar winds are < 2 ms$^{-1}$ so contribute little but spatial variability clearly makes a significant contribution for Mie winds, especially at MARA.

## 1 Introduction

The Aeolus satellite mission is the first attempt to measure meteorological wind profiles on a global scale using the Doppler lidar technique. It carries a single instrument - the Atmospheric Laser Doppler Instrument (ALADIN) - which uses two detectors to measure backscattered laser light from cloud/aerosol particles (Mie scatter) and molecules (Rayleigh

scatter), respectively (Stoffelen et al.,2005; ESA, 2008; Reitbuch, 2012). It was launched on 22 August 2018 and, from the planning stage, a wide range of validation tests were proposed, comparing the wind profiles from the satellite with those measured by established techniques such as radiosondes, ground-based radars and lidars.

Validation exercises soon after the start of the mission found that the quality of retrieved winds in part depended on the satel-lite's geolocation and on orbit orientation (see e.g. Guo et al, 2021; Lux et al., 2021). This could be traced back to unex-

pected instrumental effects, most prominently the influence of temperature on the performance of the primary telescope mir-ror of the instrument (Witschas et al., 2020; Lux et al., 2021; Weiler et al., 2021). The subsequent changes to the data-pro-cessing gave substantial improvement of the biases from more than 5 ms$^{-1}$ (Martin et al., 2021; Rennie and Isaksen, 2020) to less than 2 ms$^{-1}$ (e.g. Iwai et al., 2021; Baars et al., 2022). However, Baars et al. (2022) noted that those improvements were partly masked by worsening instrument performance (e.g. decrease in laser output energy) that led to an increase of the ran-

dom error. Nevertheless, Aeolus winds have been shown to make a positive contribution to global weather forecasting (Reit-ebuch et al,. 2020; Rennie et al., 2021; Weiler at al., 2021). A good number of validation comparisons of the corrected data processing after 2020 against a variety of other data sources have been reported, such as radiosondes (e.g. Martin et al. 2021; Rani et al., 2022 ; Chou et al., 2022), wind profiling radar (e.g. Guo et al, 2021; Kottayil et al., 2022; Chou et al., 2022), Doppler wind lidars (e.g. Chen et al., 2022, Witschas et al., 2022), numerical weather prediction models (e.g. Lux et al.,

2022, Rani et al., 2022), and other satellites (Lukens et al., 2022). Overviews of recent validation comparisons were sum-marised by e.g. Wu et al (2022) and Ratynski et al. (2023), which mostly indicate possible biases less than 1 ms$^{-1}$ and ran-dom errors 4-7 ms$^{-1}$ for Rayleigh winds, 2-4 ms$^{-1}$ for Mie winds. At the same time, the biases and random errors seem to vary more than might be expected between the different measurement techniques and locations used in the validations. Lux et al. (2022) have looked in detail at the non-random nature of differences between Aeolus winds and reference winds and suggest

that the exact details of quality control applied in validation studies can significantly affect the results. They found that the bias and random error estimates can be affected by small numbers of outliers, particularly for Mie winds where large errors outside a Gaussian distribution (gross errors) can be caused by misinterpretation of noise as signal. This can lead to predomi-nantly positively-biased gross errors.

An initial validation comparing measurements from two wind-profiler radars in Arctic Sweden and in Antarctica, with Aeo-

lus winds processed with the 2B10 baseline, was published by Belova et al. (2021a). This found biases < 2 ms$^{-1}$, and standard deviation of the differences between satellite and radar winds in the range 4-7 ms$^{-1}$. Note that a large bias first reported for Mie winds in the data for Antarctica was found to be in error as detailed in the Corrigendum published in May 2022 (Belova

et al., 2021a, Corrigendum). However, the available time period for comparison was short (only 6 months) and uncertainties in the biases were large. Almost 2 years of data from these high-latitude radars are now available for comparison with the

longest available consistently processed Aeolus data set (baseline 2B11), from July 2019 to May 2021. A comparison of these extended data sets, together with more detailed consideration of the statistics as suggested by Lux et al. (2022) is presented here.

## 2 Overview of measurements and quality criteria

The radars used are MARA (Moveable Atmospheric Radar for Antarctica), situated at Maitri in Antarctica (70.77° S, 11.73°E) and ESRAD (ESrange atmospheric RADar), situated near Kiruna in Arctic Sweden (67.88° N, 21.10° E). Full details of the radar and satellite operation modes and the available data can be found in Belova et al. (2021a,b). Each radar measures profiles of vertical and horizontal wind components in the vertical direction above the radar site. They switch automatically every 1-2 minutes between different modes with vertical resolution of 75 m, 150 m and 600 m (MARA)/ 900 m

(ESRAD). The radars sample a cone of the atmosphere with a width of about 5° for ESRAD, 10° for MARA, so the horizontal diameter of the radar beams in the lowest 10 km of the atmosphere is less than 2 km at MARA, 1 km at ESRAD. Random errors (standard deviation of all 1 or 2-minute estimates in the 1-hour averages) are typically 2-3 ms$^{-1}$ for both radars (Belova et al., 2021b). Comparison with radiosondes (Belova et al., 2021b). has shown no significant bias (<0.25 ms$^{-1}$) for winds at MARA but systematic biases at ESRAD of 8% for zonal winds, 25% for meridional winds (ESRAD underestimates wind

components). These are thought to be due to the geometry of the radar antenna field and a high level of local radio noise. The ESRAD wind estimates are corrected for these biases before being compared with Aeolus winds. For the comparison with Aeolus (as in Belova et al., 2021a), we use 1-hour averaged winds, also averaged over the height intervals corresponding to the Aeolus Rayleigh wind averages. We use only radar measurements where the 95% confidence limit of the 1-h mean is less than 2 ms$^{-1}$ (this is calculated from the standard deviation and the number of the samples in the 1-h average, using stu-

dents t-test).

We select all satellite measurement tracks passing within 100 km from each radar site. For Aeolus Rayleigh (clear) winds, we then select the profile with the mean position closest to the radar (which is averaged over about 87 km along the track). For Aeolus Mie (cloudy) winds, which are averaged over about 14 km of track, we collect all observations within 100 km of the radar and average them within the same height bins as the corresponding Rayleigh profile. We use the horizontal line-of-

sight ('HLOS') winds from the Level_2B data-product, here using the 2B11 baseline. Radar measurements of the full wind vector are averaged from 30 minutes before the pass, to 30 minutes after the pass, again to the same height bins as the Rayleigh wind profile. Radar 'HLOS' winds are calculated from the radar vector winds (ignoring the vertical component,

which is found to be negligible in the 1 h averages). There are usually 4 Aeolus passes per week providing comparative data at MARA, 3 passes per week at ESRAD.

For the analysis in Belova et al. (2021a) only winds less than 100 m/s (radar and Aeolus), validity flag =1 (Aeolus) and with estimated error (EE, also included in the Aeolus Level 2B product) < 8 ms$^{-1}$ (Rayleigh), EE < 5 ms$^{-1}$ (Mie), and 95% confidence limit < 2 ms$^{-1}$ (radar) were used. Here, as suggested by Lux et al. (2022), we first examine the statistics of the differences between radar and Aeolus winds for different quality criteria (QC). Differences are parameterised in terms of bias, standard deviation (SD), scaled median absolute deviation (ScMAD), where

$$bias = \frac{1}{N} \cdot \sum_{i=1}^{N} \left( HLOS_{Aeolus,i} - HLOS_{radar,i} \right) \tag{1}$$

$$SD = \sqrt{\frac{1}{N-1} \sum \left( \left( HLOS_{Aeolus,i} - HLOS_{radar,i} \right) - bias \right)^2} \tag{2}$$

$$ScMAD = 1.4826 \cdot median \left( \left| \left( HLOS_{Aeolus,i} - HLOS_{radar,i} \right) - median \left( HLOS_{Aeolus,i} - HLOS_{radar,i} \right) \right| \right) \tag{3}$$

Both SD and ScMAD are estimates of the variability of the wind error but ScMAD is less susceptible to outliers. If the distri-
bution is Gaussian, they have the same value.

In order to determine suitable QC, we first look at these parameters as a function of EE threshold for Rayleigh and Mie winds, and with and without a second QC, designed to eliminate gross errors, based on the modified Z-score (ModZ) (Iglewicz and Haglin, 1993), as suggested by Lux et al (2022) :

$$ModZ_i = \frac{\left| \left( HLOS_{Aeolus,i} - HLOS_{radar,i} \right) - median \left( HLOS_{Aeolus,i} - HLOS_{radar,i} \right) \right|}{ScMAD} \tag{4}$$

Figure 1 shows the fraction of possible comparison points (n) retained, biases, SD and ScMAD as a function of the EE threshold used for rejection, at MARA. Parameters with subscript z ($n_z$, $bias_z$, $SD_z$, $ScMAD_z$ ) have been calculated after further rejecting data points with $ModZ_i > 3.5$. Values for this limit between 3.0 and 3.5 were found to lead to a high degree of

15 4

normality for differences between Aeolus observations and ECMWF background winds by Lux et al. (2022). We have also tested rejecting $ModZ_i > 3.0$, but the differences are very small so we show only results using $ModZ_i > 3.5$.

In Figure 1, for Rayleigh winds it is clear that SD rises steeply for EE>7 ms$^{-1}$ but this is much less apparent where the check on $ModZ_i$ has removed outliers ($SD_z$). ScMAD is insensitive to the $ModZ_i$ restriction and is close to $SD_z$ up to 8 ms$^{-1}$ suggesting a close to Gaussian distribution after the $ModZ_i$ restriction. Bias and bias$_z$ are consistently small (about -0.5 ms$^{-1}$)
from EE>3.5 ms$^{-1}$ up to 8 m/s and bias$_z$ remains at this level for all EE thresholds tested. Thus, the original choice of EE<8 ms$^{-1}$ as the QC for Rayleigh winds seems reasonable. For Mie winds at MARA, both SD and bias increase sharply for EE>5 ms$^{-1}$. ScMAD$_z$ and SD$_z$ remain very close to each other up to EE<8.5ms$^{-1}$. Bias$_z$ remains small and at rather constant level from EE<5 to EE<8.5 ms$^{-1}$. The fraction of total comparison points left after applying both the EE and $ModZ_i$ rejection criteria ($n_z$) increases sharply for EE < 5 ms$^{-1}$ and more slowly after that to just over 80% for Rayleigh (corresponding to
~800 points) and to about 70% for Mie winds (~350 points). So, in order to include as many points as possible and a distribution as close as possible to Gaussian, it seems reasonable to increase the original threshold of EE<5 ms$^{-1}$ for Mie winds, to anywhere up to EE<8 ms$^{-1}$ together with the outlier rejection using $ModZ_i < 3.5$.

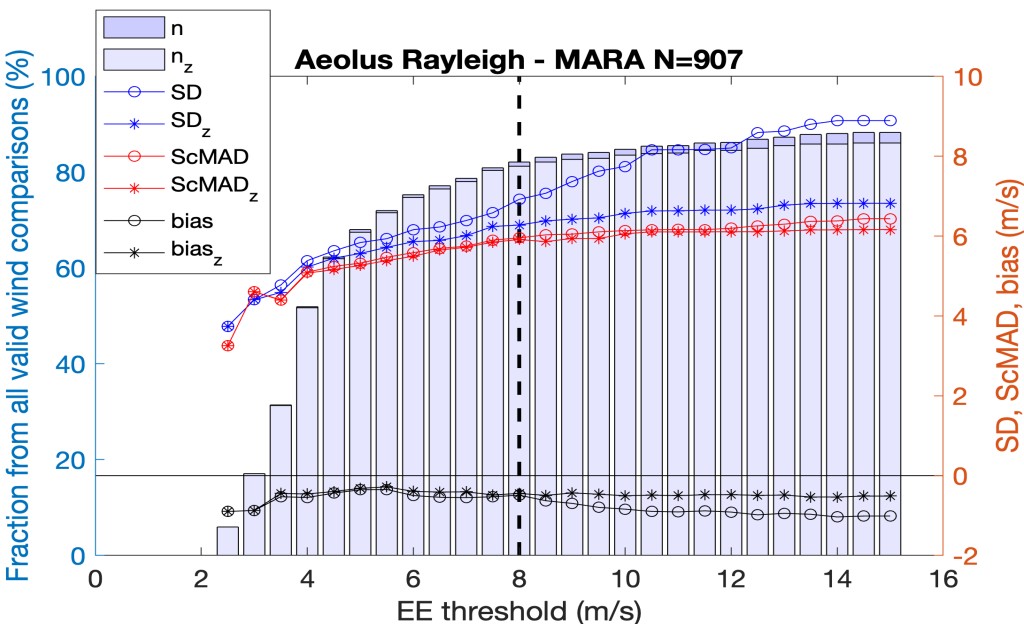

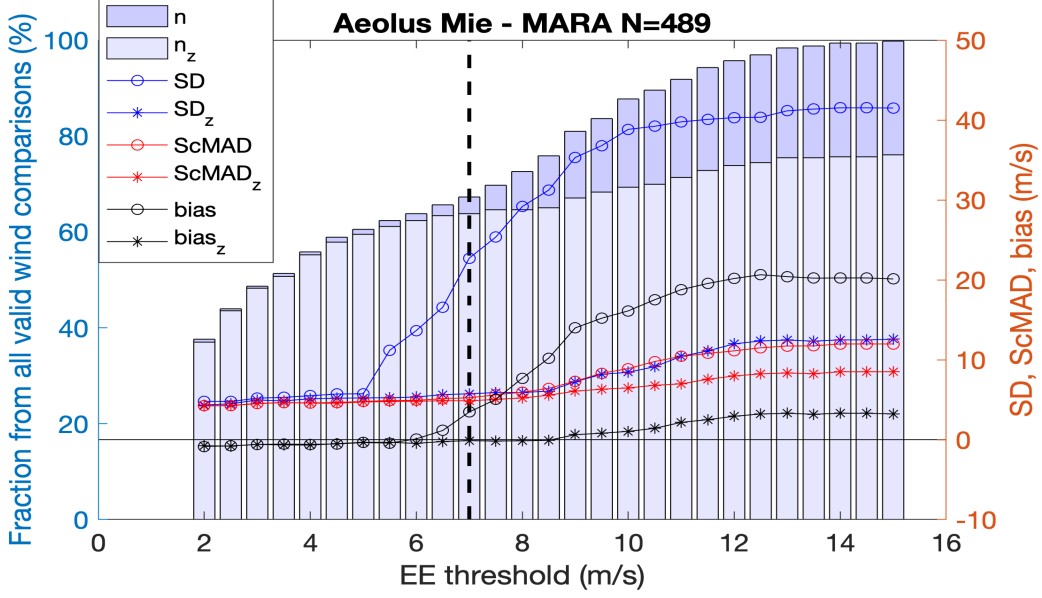

**Figure 1: Comparison of Aeolus HLOS winds with MARA, for all available data, upper panel for Rayleigh-clear winds, lower panel for Mie-cloudy. Plots show the fraction of possible comparison points (n) retained, biases, SD and ScMAD as a function of the EE threshold used for rejection. Subscript z ($n_z$, $bias_z$, $SD_z$, $ScMAD_z$ ) show results after further rejecting data points with ModZ > 3.5. N in the panel titles is the number of samples corresponding to n=100%. The vertical dashed line marks the EE threshold used for the analysis in the rest of the paper.**


Figure 2 shows corresponding plots at ESRAD. For Rayleigh winds, the bias and $bias_z$ are insensitive to the EE threshold from EE<5 ms$^{-1}$ to EE<15 ms$^{-1}$. SD increases sharply at EE<10 ms$^{-1}$ and above. Otherwise, $SD_z$, ScMAD , $ScMAD_z$ and the difference between $SD_z$ and $ScMAD_z$ all increase slowly but steadily for all EE thresholds above EE<5 ms$^{-1}$. Thus, there is no clear motivation for a particular choice of EE threshold for Rayleigh winds. For Mie winds at ESRAD, SD and bias in-
crease rapidly with EE thresholds > 5 ms$^{-1}$ while $SD_z$, ScMAD , $ScMAD_z$ and the difference between $SD_z$ and $ScMAD_z$ seem to increase more rapidly for EE threshold > 7.0 ms$^{-1}$. The $bias_z$ increases slowly across all EE thresholds but is fairly constant between EE<5 ms$^{-1}$ and EE<7 ms$^{-1}$. The fraction of total comparison points left after applying both the EE and $ModZ_i$ rejection criteria ($n_z$) increases sharply for EE < 5 ms$^{-1}$ and more slowly after that to just over 90% for Rayleigh (correspond-
ing to ~1800 points) and to about 70% for Mie winds (~700 points).

Figures 1 and 2 show very similar behaviour at MARA and at ESRAD so there is no obvious reason to treat the data from the two sites differently. We have made similar plots for all of the data subsets which we analyse below and found no reason to choose different thresholds for the different subsets. In all cases, ScMAD is close to $ScMAD_z$ and their values are constant or changing very slowly between EE values 1 ms$^{-1}$ above or below the thresholds. Similarly, bias and $bias_z$ are close together and insensitive to the EE values around the chosen thresholds, although both the $bias_z$ and $ScMAD_z$ values can lie at different
levels in the different subsets, as shown in Tables 1 and 2 and discussed in the next section. Thus, in the following we adopt QC using only Aeolus winds with estimated random error (EE) < 8 ms$^{-1}$ (Rayleigh), < 7 ms$^{-1}$ (Mie) and rejecting likely gross errors where ModZ > 3.5. This results in 80-90% of Rayleigh wind comparison points and about 60% of Mie wind points being available for analysis, sufficient numbers for further division according to summer/winter and ascending/descending orbits. The same restrictions on radar winds as in Belova et al., 2021a are also applied - wind speed less than 100 ms$^{-1}$ and
95% confidence limit < 2 ms$^{-1}$.



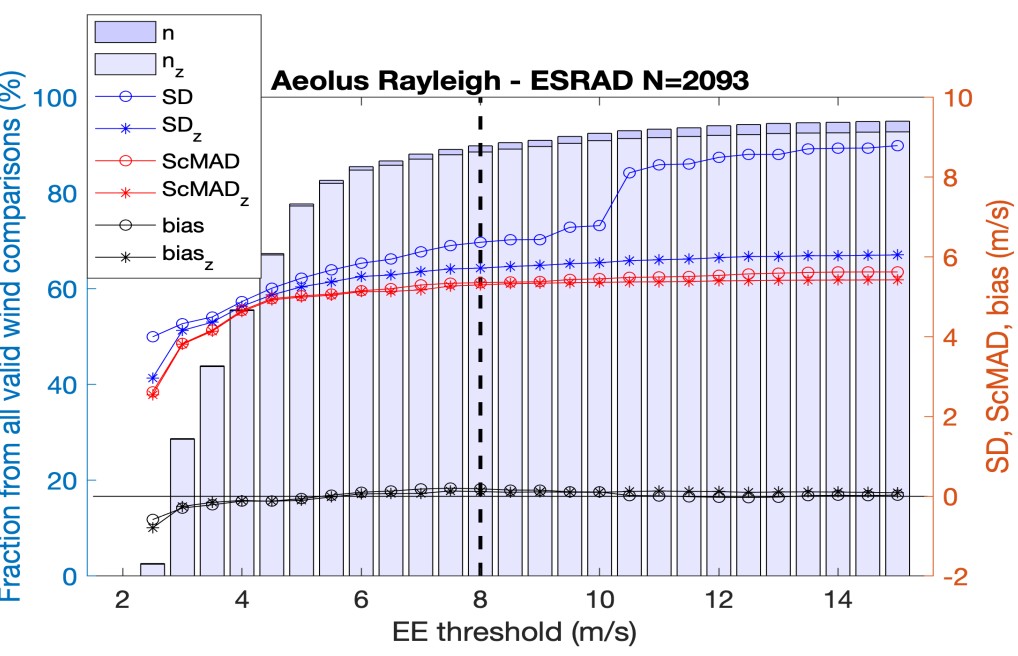



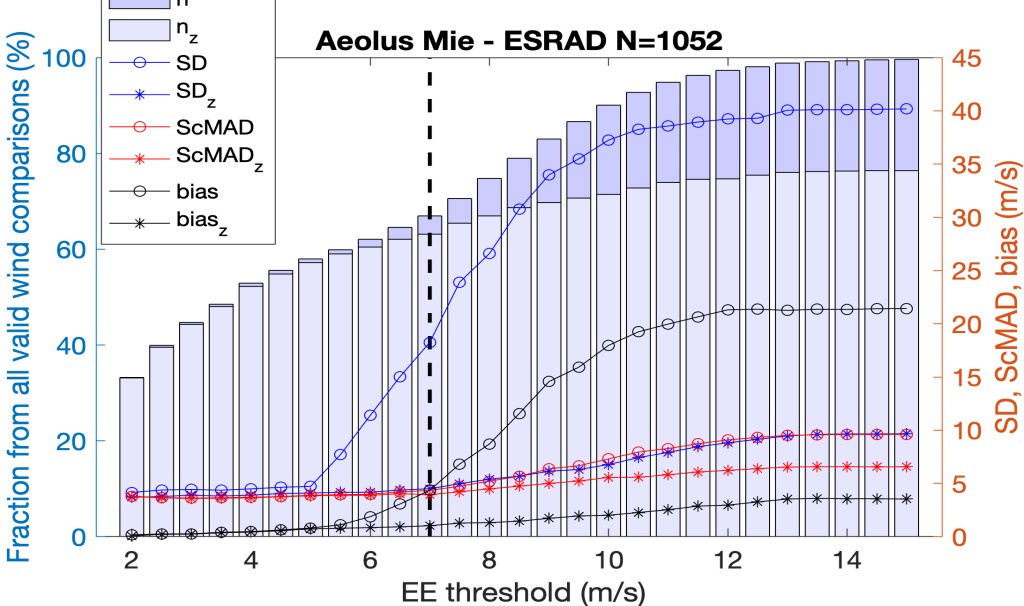

**Figure 2: As Figure 1 but for ESRAD.**

**Table 1: Statistics of correlation and differences between Aeolus Rayleigh-clear HLOS winds and MARA HLOS winds. | u$_{HLOS}$ | shows the median Aeolus HLOS wind speed in each data subset, with the values between square brackets [] corresponding to the**
**lower and upper quartiles of the distribution. N$_z$ is the number of comparison points passing all quality checks (QC, see text for details), % outliers is the number of points rejected by the final QC (ModZ<3.5, Eq. 4). Slope$_z$ is the slope of the best-fit straight line correlation, bias$_z$, SD$_z$ and ScMAD$_z$ are as defined in Eqs. 1-3. Columns are for all data (July 2019 - May 2021), or divided into summer (23 September-22 March), winter (23 March-22 September), descending and ascending passes. For slope$_z$ and bias$_z$ , values between square brackets [] are 95% confidence limits. Rayleigh winds with EE>8 ms$^{-1}$ are excluded.**

| Rayleigh-MARA | all | SH summer | SH winter | descending all seasons | ascending all seasons | within 25 km all seasons descending |
|---|---|---|---|---|---|---|
| \| u$_{HLOS}$\|  ms$^{-1}$ | 7 [ 3 11 ] | 7 [ 3 11 ] | 6 [ 3 11 ] | 7 [ 3 11 ] | 7 [ 3 11 ] | 7 [ 3 11 ] |
| N$_Z$ | 737 | 553 | 294 | 351 | 387 | 211 |
| % outliers | 1.1 | 1.1 | 1.0 | 0.8 | 1.0 | 0.5 |
| correlation$_Z$ | 0.77 | 0.78 | 0.77 | 0.68 | 0.77 | 0.67 |
| slope$_Z$ | 0.90 [0.84 0.95] | 0.92 [0.85 0.99] | 0.89 [0.80 0.97] | 0.93 [0.82 1.03] | 0.98 [0.89 1.05] | 0.88 [0.74 1.02] |
| bias$_Z$ ms$^{-1}$ | -0.5 [ -0.9 0.0] | 0.0 [-0.6 0.6] | -0.8 [-1.6 -0.1] | -1.4 [-2.1 -0.7] | 0.6 0.0  1.1] | -1.4 [-2.3 -0.5] |
| SD$_Z$ ms$^{-1}$ | 6.3 | 6.2 | 6.3 | 6.8 | 5.8 | 6.9 |
| ScMAD$_Z$ ms$^{-1}$ | 5.9 | 5.8 | 5.9 | 6.8 | 5.5 | 6.7 |


35

**Table 2: As Table 1, but for Aeolus Mie-cloudy HLOS winds and MARA HLOS winds. Mie winds with EE>7 ms⁻¹ are excluded.**

| Mie-MARA | all | SH summer | SH winter | descending all seasons | ascending all seasons | within 25 km all seasons descending |
|---|---|---|---|---|---|---|
| $\lvert u_{HLOS}\rvert$ ms⁻¹ | 8 [ 4 13 ] | 8 [ 4 12 ] | 7 [ 4 15 ] | 8 [ 3 11 ] | 8 [ 5 15 ] | 9 [ 6 13 ] |
| $N_z$ | 312 | 208 | 102 | 146 | 165 | 66 |
| % outliers | 5.2 | 4.1 | 8.9 | 2.7 | 7.8 | 2.9 |
| correlations | 0.86 | 0.87 | 0.84 | 0.78 | 0.70 | 0.86 |
| $slope_z$ | 0.96 [0.89 1.01] | 0.90 [0.83 0.96] | 1.11 [0.97 1.25] | 0.94 [0.81 1.06] | 0.99 [0.84 1.13] | 0.99 [0.84 1.13] |
| $bias_z$ ms⁻¹ | -0.1 [-0.8 0.5] | 0.1 [-0.6 0.8] | -0.6 [-1.9 0.7] | -0.4 [-1.3 0.4] | -0.2 [-1.1 0.8] | 0.0 [-0.8 0.9] |
| $SD_z$ ms⁻¹ | 5.7 | 5.0 | 6.6 | 5.1 | 6.2 | 3.5 |
| $ScMAD_z$ ms⁻¹ | 4.9 | 4.6 | 6.4 | 4.2 | 5.3 | 3.3 |

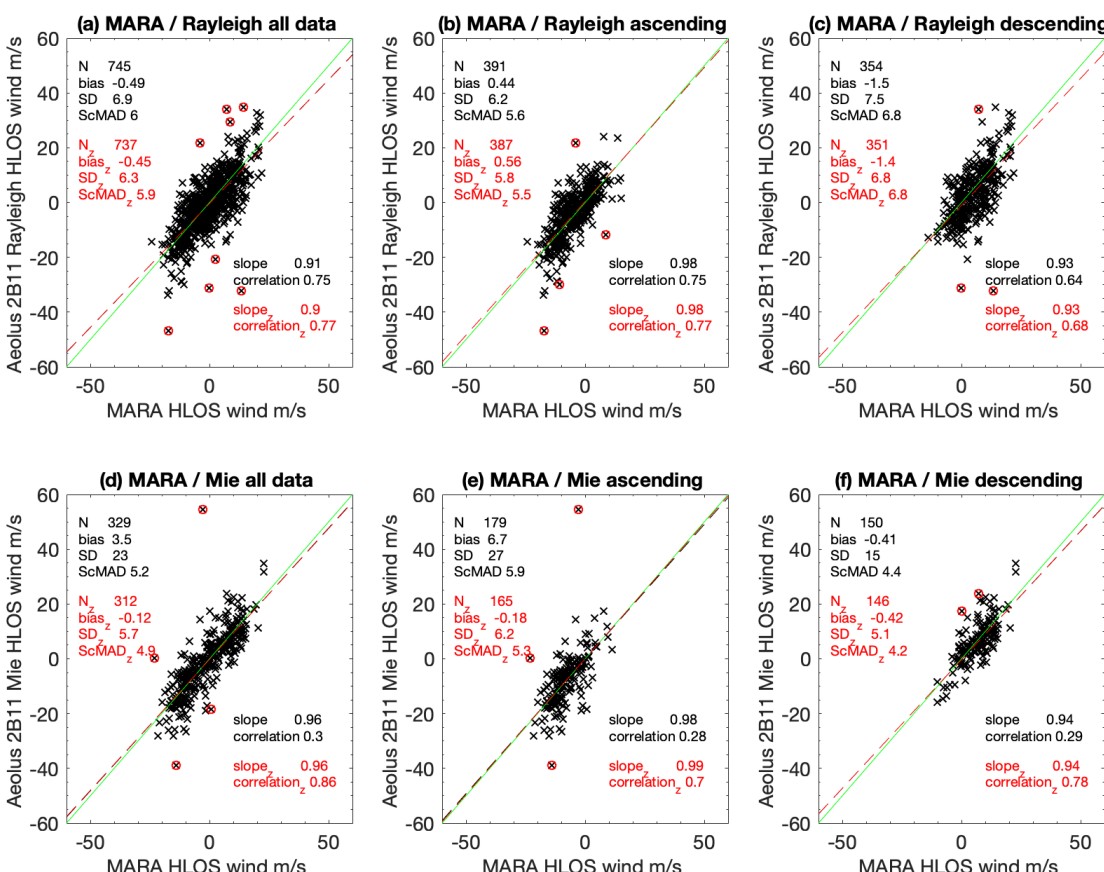

Figure 3: Scatter plots of Aeolus HLOS Rayleigh-clear winds (a-c) and Mie-cloudy winds (d-f) vs MARA winds. (a),(d) show all orbits together, (b),(e) show ascending passes and (c) ,(f) descending passes. Red circles show data points removed by the ModZ>3.5 QC criterium. Parameters in black/red indicate fits including/excluding these points. Green line is where Aeolus wind is exactly equal to MARA wind. Units for bias$_z$, SD$_z$ and ScMAD$_z$ are ms$^{-1}$.

## 3 Comparison with MARA radar, Antarctica

Statistics of the comparison between Aeolus and MARA are given in Table 1 (Rayleigh winds) and Table 2 (Mie winds), and scatter plots of the comparisons are shown in Figure 3. (Note that no correction is made in the Tables for the random

uncertainties in radar measurements). The first column in each table shows the comparison for all of the data, corresponding to Fig. 1 at EE thresholds 8 ms$^{-1}$ for Rayleigh winds, 7 ms$^{-1}$ for Mie winds. The tables also show the results after dividing the data by season (summer, 23 September - 22 March and winter 23 March - 22 September) and by ascending (afternoon) / descending (morning) Aeolus passes. Variations of, for example, solar illumination on the ground between summer and winter and opposite lidar backscatter direction relative to the prevailing wind between ascending and descending passes could in

principle affect the comparison. Seasonal influences on the instrument performance have also been found to be important, particularly for the bias (Weiler et al., 2021). Tables 1 and 2 also include a further column which shows the results when the comparison is restricted to Aeolus measurements within 25 km of MARA (more precisely, those with the mid-point of the average along the orbit track within 25 km). Because of the geometry of the satellite orbit, these are all on descending passes. Note that since Rayleigh winds are averaged over about 87 km distance along the track, those measurements will still include

observations up to 68 km from the radar along the track. For Mie winds, which are averaged over 15 km, observations up to 33 km away can contribute.

For the Rayleigh winds, Table 1 shows that there are no significant differences between summer and winter (Note that results restricted to Aeolus measurements only within 25 km from the radar are shown only in the Tables. All of the figures include points up to 100 km from the radar). There does seem to be a significant difference between ascending and descending

passes with descending passes showing lower correlation, stronger (negative) bias and higher SD$_z$ and ScMAD$_z$. These differences can also be discerned comparing Figs 3b and 3c. For all of the data together, there is a small, marginally significant bias. As can be seen in Fig.1, this bias is largely independent of the choice of EE threshold for data rejection.

For the Mie winds, Table 1 shows that there are (in %) twice as many outliers rejected by the ModZ<3.5 QC in winter compared to summer, SD$_z$ and ScMAD$_z$ are also higher in winter, but the biases are not significantly different. Ascending passes

show a higher rate of outliers, and higher SD$_z$/ScMAD$_z$ compared to descending, but again with no significant bias for either. Again the differences can be discerned comparing Figs 3e and 3f. The differences in variability between ascending and descending passes are opposite for Mie winds compared to Rayleigh winds, the differences in variability between summer and winter affect only the Mie winds and significant bias for the descending passes affects only the Rayleigh winds, so they are unlikely to be explained by meteorology or by systematic errors in radar wind speed. Overall, SD$_z$ and ScMAD$_z$ are slightly

higher for Rayleigh winds (around 6 ms$^{-1}$) than for Mie winds (around 5 ms$^{-1}$). Comparing the red and black numbers in Fig-

ure 3 also shows the large change in SD and bias for Mie winds when the ModZ<3.5 QC is applied (comparing bias/SD with bias$_z$/SD$_z$).

Figure 4 shows height resolved parameters for the Aeolus-MARA comparison. Fig. 4a and 4d shows that low heights between 1-5 km dominate the comparison even though  Aeolus  wind estimates are available throughout the troposphere (and higher in the case of Rayleigh winds). This is due to the low sensitivity of the MARA radar in the upper troposhere and above. The uncertainty in radar winds is shown by the green line in  Fig. 4b and 4c. Each radar wind is estimated from a 1-h average of measurements and the standard error of the mean (SEM) is used as an estimate of the uncertainty. Since we include only averaged radar winds where the 95% confidence interval is $< 2$ ms$^{-1}$ (this is twice the SEM when the number of data points in the average is large) , SEM is low, below 1 ms$^{-1}$ and increases only slightly with height. (The SEM$_{MARA}$ profile is essentially the same for the ascending and descending passes as for all data so, for clarity, it is not included in the plot.) In Figs. 4e and 4f we can see that the negative bias for Rayleigh descending winds, seen in Tables 1 and 2, is seen at almost all heights, although the uncertainties in the bias become very large above 6 km height.  It is partly balanced by a positive bias (marginally significant) for the ascending passes so that, for all data together (Figs. 4b and 4e), the mean bias becomes closer to zero. For the Mie winds, with notably more restricted height coverage, there is no significant bias at any height.


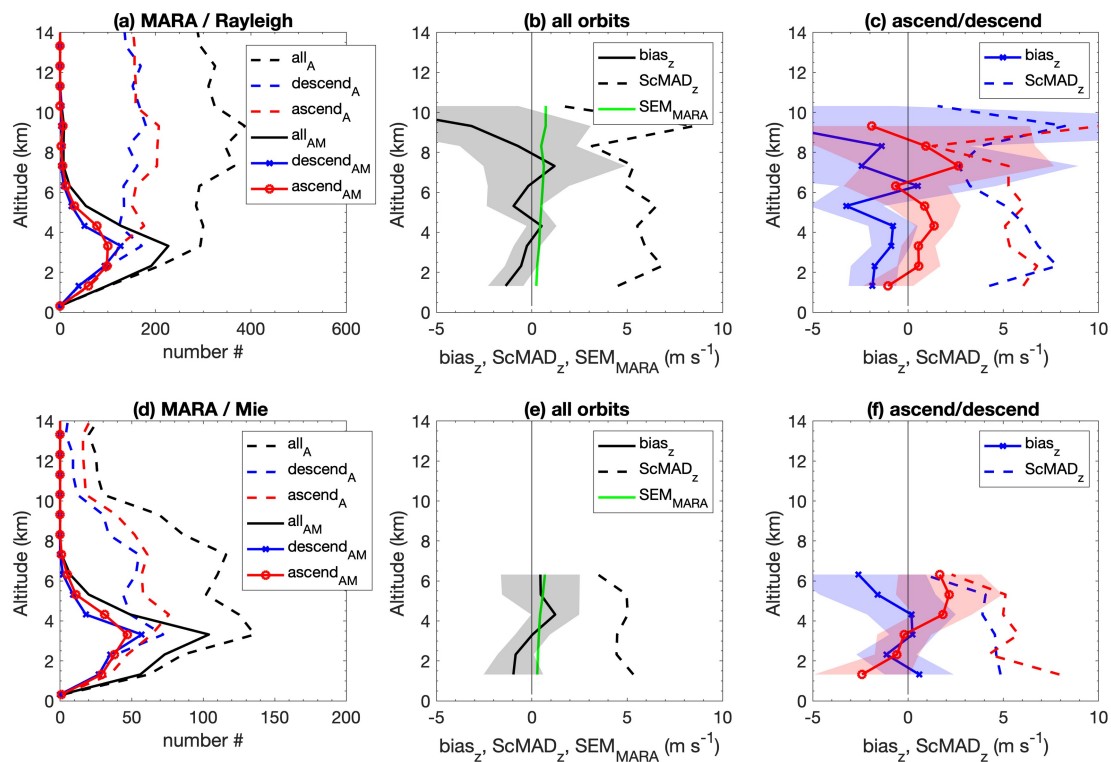

**Figure 4: Comparison of Aeolus winds with MARA. Panels (a),(d) show height profiles of numbers of data points, dashed limes, subscript A, show the number of Aeolus wind observations with EE>8ms-1 (Rayleigh) or 7 ms-1 (Mie). Solid lines, subscript AM , shows the number of points included in the analysis, i.e. where MARA data is also available and modZ <3.5.  Panels (b),(e) show height profiles of the mean values of the uncertainty in MARA wind estimates (green line, $SEM_{MARA}$ ),  $bias_z$  and $ScMAD_z$  for all orbits together. Panels  (c),(f) show $bias_z$ and $ScMAD_z$ separately for ascending and descending passes.**



**Table 3: Statistics of correlation and differences between Aeolus Rayleigh-clear HLOS winds and ESRAD HLOS winds. | u $_{HLOS}$ | shows the median Aeolus HLOS wind speed in each data subset, with the values between square brackets [] corresponding to the lower and upper quartiles of the distribution. N$_z$ is the number of comparison points passing all quality checks (QC, see text for details), % outliers is the number of points rejected by the final QC (ModZ<3.5, Eq. 4). Slope$_z$ is the slope of the best-fit straight line correlation, bias$_z$, SD$_z$ and ScMAD$_z$ are as defined in Eqs. 1-3. Units for bias$_z$, SD$_z$ and ScMAD$_z$ are ms$^{-1}$. Columns are for all data (July 2019 - May 2021), or divided into summer (23 March-22 September), winter (23 September-22 March), descending and ascending passes. For slope$_z$ and bias$_z$, values between square brackets [] are 95% confidence limits. Rayleigh winds with EE>8 ms$^{-1}$ are excluded.**

| Rayleigh-ESRAD | all | NH summer | NH winter | descending all seasons | ascending all seasons | within 25 km all seasons ascending |
|---|---|---|---|---|---|---|
| $\mid u_{HLOS} \mid$ ms$^{-1}$ | 7 [ 3 13 ] | 7 [ 3 12 ] | 8 [ 84 13 ] | 7 [ 3 13 ] | 7 [ 3 13 ] | 7 [ 3 13 ] |
| N$_Z$ | 1854 | 959 | 895 | 1220 | 634 | 624 |
| % outliers | 1.4 | 1.0 | 1.9 | 1.2 | 1.9 | 1.9 |
| correlation$_Z$ | 0.87 | 0.84 | 0.88 | 0.82 | 0.83 | 0.81 |
| slope$_Z$ | 0.99 [0.96 1.01] | 1.00 [0.96 1.04] | 0.96 [0.93 1.00] | 0.97 [0.93 1.00] | 0.96 [0.91 1.01] | 0.95 [0.90 1.01] |
| bias$_Z$ ms$^{-1}$ | 0.1 [-0.1 0.4] | 0.3 [-0.1 0.7] | -0.1 [-0.4 0.3] | -0.0 [-0.4 0.3] | 0.4 [0.0 0.9] | 0.4 [0.0 0.9] |
| SD$_Z$ ms$^{-1}$ | 5.7 | 5.9 | 5.5 | 5.8 | 5.6 | 5.7 |
| ScMAD$_Z$ ms$^{-1}$ | 5.3 | 5.4 | 5.1 | 5.2 | 5.2 | 5.3 |

**Table 4: As Table 3, but for Aeolus Mie-cloudy HLOS winds and ESRAD HLOS winds. Mie winds with EE>7 ms$^{-1}$ are excluded.**

| Mie-ESRAD | all | NH summer | NH winter | descending all seasons | ascending all seasons | within 25 km all seasons ascending |
|---|---|---|---|---|---|---|
| $\lvert u_{HLOS} \rvert$ ms$^{-1}$ | 6 [ 3 12 ] | 5 [ 2 10 ] | 7 [ 4 14 ] | 6 [ 3 11 ] | 7 [ 3 14 ] | 7 [ 3 15 ] |
| $N_Z$ | 661 | 362 | 300 | 402 | 259 | 140 |
| % outliers | 5.7 | 3.7 | 3.0 | 4.0 | 2.6 | 5.4 |
| correlation$_Z$ | 0.91 | 0.89 | 0.92 | 0.86 | 0.89 | 0.89 |
| slope$_Z$ | 0.94 [0.91 0.97] | 0.85 [0.80 0.89] | 1.03 [0.98 1.07] | 0.92 [0.86 0.97] | 0.93 [0.87 0.98] | 0.98 [0.90 1.07] |
| bias$_Z$ ms$^{-1}$ | 1.0 [0.7 1.4] | 1.1 [0.6 1.5] | 1.0 [0.5 1.6] | 1.1 [0.6 1.6] | 0.9 [0.4 1.4] | 0.7 [0.0 1.4] |
| SD$_Z$ ms$^{-1}$ | 4.5 | 4.3 | 4.8 | 4.8 | 4.0 | 4.2 |
| ScMAD$_Z$ ms$^{-1}$ | 3.9 | 3.9 | 4.1 | 4.0 | 4.0 | 3.6 |

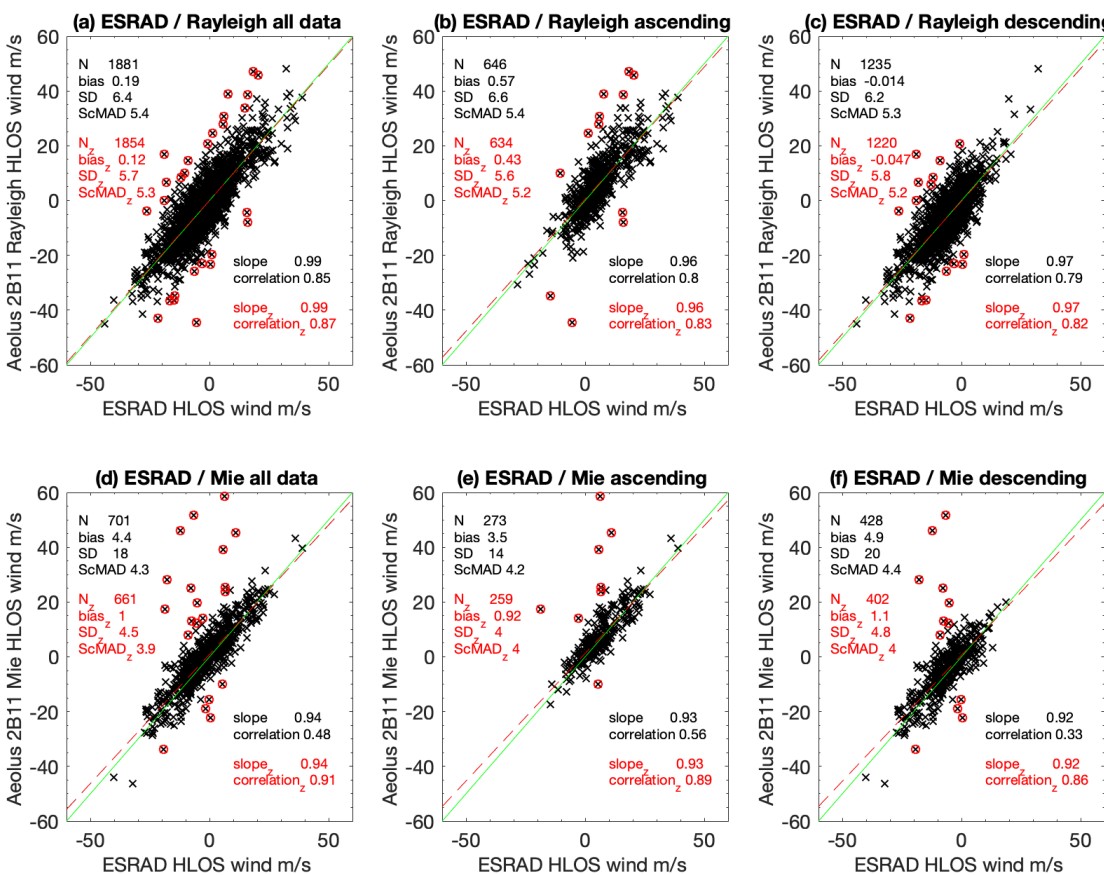

**Figure 5: As Figure 3 but for ESRAD**

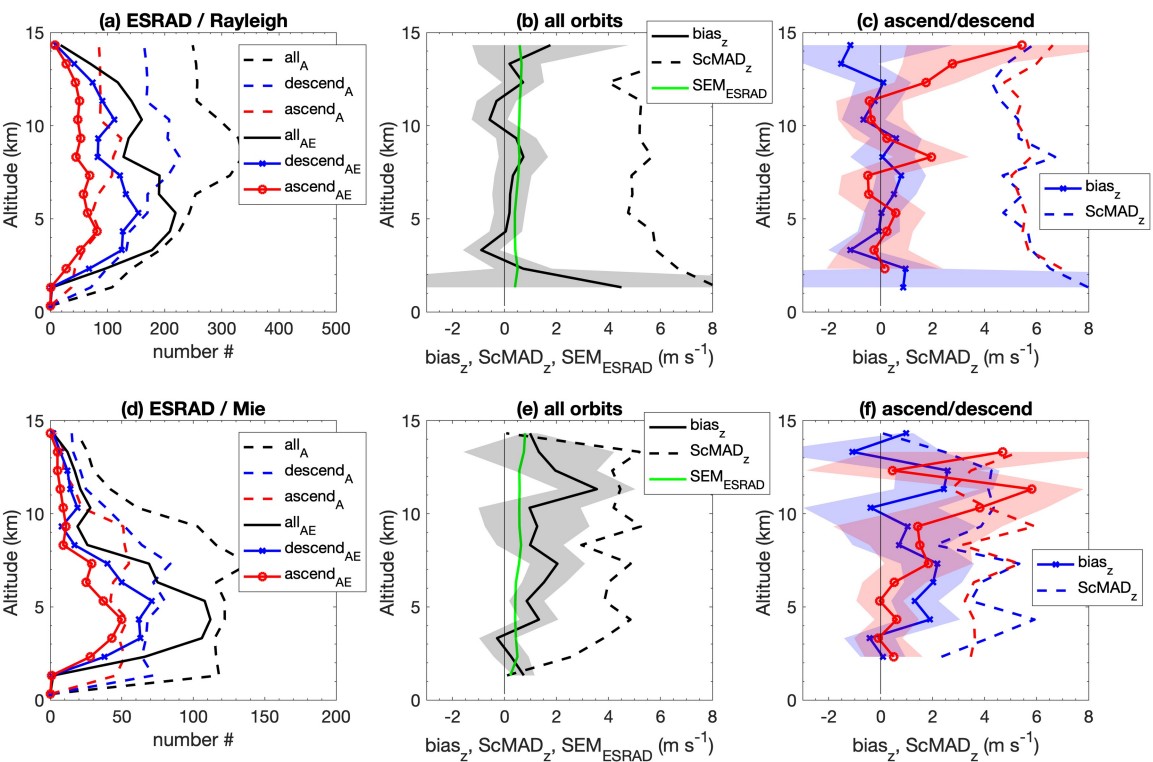

**Figure 6: Comparison of Aeolus winds with ESRAD. Panels (a),(d) show height profiles of numbers of data points, dashed limes, subscript A, show the number of Aeolus wind observations with EE>8ms-1 (Rayleigh) or 7 ms$^{-1}$ (Mie). Solid lines, subscript AE , shows the number of points included in the analysis, i.e. where ESRAD data is also available and modZ <3.5. Panels (b),(e) show height profiles of the mean values of the uncertainty in ESRAD wind estimates (green line, SEM$_{ESRAD}$ ), bias$_z$ and ScMAD$_z$ for all orbits together. Panels (c),(f) show bias$_z$ and ScMAD$_z$ separately for ascending and descending passes.**

315

## 4 Comparison with ESRAD radar, Arctic Sweden

Table 3 (Rayleigh winds) and Table 4 (Mie winds) show statistics of the comparison between Aeolus and ESRAD. Scatter plots of the comparisons are shown in Figure 5. The comparison for all of the data is shown in the first column in each table, corresponding to Fig. 2 at EE thresholds 8 ms$^{-1}$ for Rayleigh winds, 7 ms$^{-1}$ for Mie winds. The tables also show the results after dividing the data by season (winter, 23 September - 22 March and summer 23 March - 22 September) and by ascending (afternoon) / descending (morning) Aeolus passes. The final column in Tables 3 and 4 shows the results when the comparison is restricted to Aeolus measurements with their mid-points within 25 km of ESRAD. These are all on ascending passes and, due to averaging, will include observations up to 68 km /33 km from the radar along the track for Rayleigh / Mie winds respectively.

In Table 3 (Rayleigh winds), there are no significant differences between summer and winter, or between ascending and descending passes, and bias$_z$ in all cases is not significantly different from zero. In Table 4 (Mie winds), there are again no significant differences between summer and winter or between ascending and descending passes. However, there is a significant bias of about 1 ms$^{-1}$ for all cases. Overall, SD$_z$ and ScMAD$_z$ are slightly higher for Rayleigh winds (around 5 ms$^{-1}$) than for Mie winds (around 4 ms$^{-1}$), and slightly lower than at MARA.

Figure 5 illustrates the distribution of points about the regression lines and shows how the rejection of points with ModZ>3.5 has effectively eliminated several gross errors. Figure 5 (comparing the black numbers for bias/SD with the red numbers for bias$_z$/SD$_z$) also shows the large change in SD and bias for Mie winds when the ModZ<3.5 QC is applied.

Figure 6 provides height-resolved profiles of parameters for the Aeolus-ESRAD comparison. As can be seen in Figs 5a and 5d, in contrast to MARA, the more powerful ESRAD radar provides useful coverage in the upper troposphere as well as the lower troposphere. There are fewer joint ESRAD-Aeolus observations than Aeolus alone although, between 2 and 5 km height almost all Aeolus measurements have corresponding radar ones, about half higher up in the troposphere. The green line in Figs 5b and 5e shows the mean SEM for the ESRAD wind averages, reaching 1 ms$^{-1}$ at in the upper troposphere, lower at lower heights. (The SEM$_{ESRAD}$ profile is essentialy the same for the ascending and descending passes as for all data so, for clarity, it is not included in the plot.) Considering the bias profiles in Figs 4b,c,e,f , above 6 km height, the bias uncertainties are notably lower than at MARA - this is a result of a much larger number of comparison points thanks to the higher power of the ESRAD radar. For Rayleigh winds there is no significant bias at any height, for Mie winds the ~1 ms$^{-1}$ positive bias identified in Table 4 is clearly seen at all heights. From Fig. 2 it is clear that a positive bias appears whatever the EE threshold.

## 5 Further analysis of non-zero biases

The analysis above has identified significant non-zero biases in two cases - for Rayleigh winds at MARA (descending passes, $bias_z$ -1.4 ms$^{-1}$) and for all Mie winds at ESRAD ($bias_z$ +1 ms$^{-1}$). To check these further, we plot normal probability curves for a series of EE thresholds in Figures 7 and 8. These plots compare the distribution of the data (Aeolus HLOS wind - radar HLOS wind, after applying the ModZ<3.5 QC) to the normal distribution ('+'). A reference line (red) joins the first and third quartiles of the data and is projected to the ends of the data. If the sample data has a normal distribution, then the

data points appear along the reference line. Departures from the line to the right at the positive end and to the left at the negative end show 'fat tails' (more points in the tails of the data distribution than in the normal distribution). When one tail is bigger than the other the distribution is skewed.

Figure 7 (Rayleigh-descend - MARA) shows fairly symmetric, small fat tails which grow slightly as the EE threshold is increased. The bias remains the same over the range of EE thresholds. This same constant bias over all EE thresholds can be

seen for all of the Rayleigh-MARA winds in Figure 1. Figure 8 (Mie - ESRAD) shows small fairly symmetric fat tails for low values of the EE threshold but these grow large and become skewed at the higher EE thresholds, leading to an increase in the bias estimate. This is also seen in Figure 2. There is no obvious reason why the distribution is skewed, only for Mie winds, and only at ESRAD. One possibility might be local meteorology as the ESRAD area is often covered by mountain-lee-wave clouds which might affect Mie (cloudy) measurements differently to Rayleigh (clear) ones. In general, vertical

winds of up to 2 m/s can be found in the troposphere in mountain lee waves at ESRAD (Kirkwood et al., 2010). However, the horizontal wavelengths of the lee waves are only a few 10s of km and would be averaged along the Aeolus track. In the comparison data set here, 99% of the data points have vertical winds within +0.4/-0.4 m/s at ESRAD and there is no correlation between vertical wind and the difference between ESRAD and Aeolus HLOS winds. So vertical winds cannot explain the skewed distribution. Preferential locations for cloud formation within the wave wind field could affect Mie winds differ-

ently from Rayleigh winds. Extensive case studies would be needed to test this possibility.

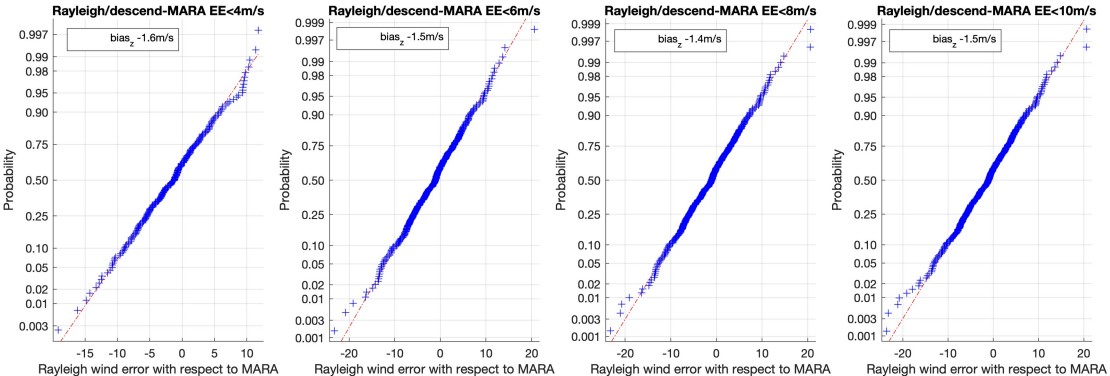

**Figure 7: Normal probability plot for the difference between Aeolus Rayleigh HLOS wind and MARA wind (descending passes) for a series of EE thresholds, after rejecting points with ModZ > 3.5. See text for details.**

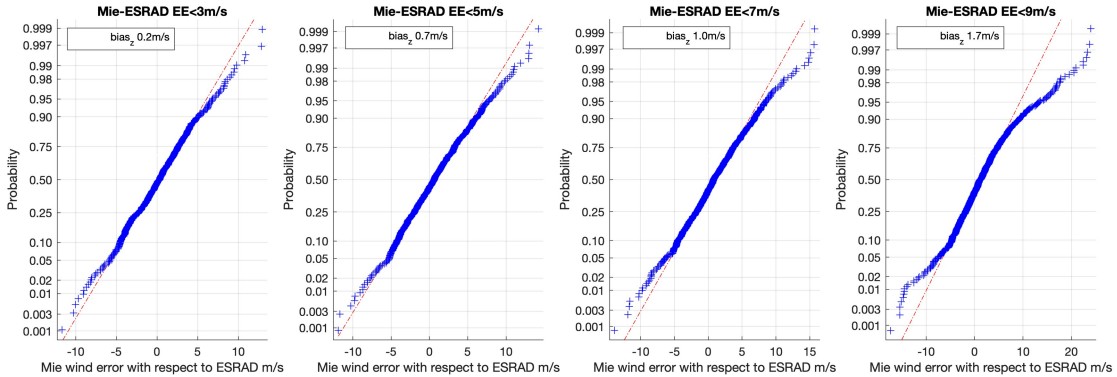

**Figure 8: Normal probability plot for the difference between Aeolus Mie HLOS wind and ESRAD wind (all passes) for a series of EE thresholds, after rejecting points with ModZ > 3.5. See text for details.**

**6 Discussion and Conclusions**

In the present study we have compared 2 years of wind measurements by the Aeolus satellite (Rayleigh-clear and Mie-cloudy) with winds from two wind-profiler radars in Arctic Sweden and in coastal Antarctica, respectively. For each radar we have looked at ascending and descending passes and summer and winter separately, as well as for all of the data together. We have identified significant non-zero biases in only two subsets of the data - for Rayleigh winds at MARA (winter de-385 scending passes, bias -1.4 ms$^{-1}$) and for Mie winds (all passes) at ESRAD (bias +1 ms$^{-1}$). Biases for all other subsets are not different from zero at the 95% confidence limits. In the initial validation of Aeolus winds against the MARA radar (Belova

et al, 2021a) significant bias (-2 ms$^{-1}$) was also found for Rayleigh winds (descending passes) at MARA which is similar to the result here. For Mie winds (Belova et al, 2021a, corrigendum), the initial study also found a positive bias similar to the present study at ESRAD (average +1.2 ms$^{-1}$). The number of comparison points has increased by about a factor 3 in the present study due the longer time period (two years instead of 6 months) and, for Mie winds, the relaxation of the random error (EE) threshold for rejection of data from 5 ms$^{-1}$ to 7 ms$^{-1}$. With the increase in numbers and the introduction of a new criterium for rejection of outliers (modZ < 3.5), the uncertainties in the bias estimates have been substantially reduced (from 1-3 ms$^{-1}$ to 0.3 – 1 ms$^{-1}$) so we can be more confident that the estimated biases are accurate. It seems clear that uncorrected biases can still appear for our particular locations even after the data processing improvements incorporated in the L2B product with 2B11 baseline. In addition, for Mie winds at ESRAD there is clearly a problem with a skewed distribution of random errors, with substantial numbers of Mie (HLOS) winds which are greater in magnitude than the radar winds, larger than expected for a normal distribution, and which are difficult to remove even with the new outlier constraint (modZ < 3.5). The problem of skewness for Mie winds has also been reported, and addressed in detail, by Lux et al. (2022).

The biases are similar in magnitude to results from other locations (e.g. Wu et al., (2022) Ratynski et al. (2023) and summaries included in those papers). Both Kottayil et al. (2022) and Ratynski et al. (2023) found no differences of the statistical results for ascending and descending passes. However, Martin et al. (2022) noted that biases can depend on latitude. It should be noted that Lukens et al. (2022) found large differences in the standard deviation of atmospheric motion vectors over Antarctica that were derived from Aeolus winds and from geostationary satellites, and indicated that this is due to problems with the correct height assignment. Chou et al. (2022) presented a validation comparison with both radiosondes and radar in northern Canada, i.e. from latitudes similar to ESRAD. They found that Aeolus winds correlate well with their radiosonde data. On the other hand, correlation with radar winds was much less good. The reasons were twofold, firstly the radar only operated for a limited time, leading to only a small number of profiles being available for comparison, secondly the range of the radar was limited, as it was not optimised to measure winds but rather hydrometeors (e.g. rain).

Random differences (ScMAD$_z$) for all data together are 5.9 / 5.3 ms$^{-1}$ for Rayleigh winds at MARA/ESRAD respectively, and 4.9 / 3.9 ms$^{-1}$ for Mie winds. We note that the random errors in the radar measurements should be < 2 ms$^{-1}$ (we have used only radar wind estimates where the 95% confidence limit for the 1-h average is < 2 ms$^{-1}$) so that this should contribute little to the SD of the differences between radar and Aeolus measurements (less than 0.5 ms$^{-1}$ assuming uncorrelated random errors). Since Aeolus HLOS winds have been sampled within a few 10s of km from the radar sites, we can use the comparison of radar measurements with radiosondes by Belova et al., 2021a to give some indication of the possible combined effects of spatial variability and random errors in the radar measurements (sondes, although launched at the radar sites, can be several 10s of km away by the time they leave the troposphere). Belova et al., 2021b found the standard deviation of differences between winds measured by the radars and sondes to be about 4 ms$^{-1}$ at MARA (covering 291 sondes between February and October 2014), 5 ms$^{-1}$ at ESRAD (28 radiosondes between January 2017 and August 2019). These are comparable to the values of ScMAD$_z$ found in the Aeolus-radar comparison for Mie winds, and slightly less than found for Rayleigh winds. So

it is clear that part of $ScMAD_z$ is likely due to spatial variability but it is not possible to accurately quantify this. Assuming that the levels found in the radiosonde comparison are representative, spatial variability could in principle account for all of $ScMAD_z$ (e.g. for the ESRAD-Mie wind comparison), or as little as 25% (e.g. for the MARA-Rayleigh wind comparison). An alternative is to consider the effect on $ScMAD_z$ of restricting the comparison to Aeolus wind measurements closer to the radars. These results are shown in the rightmost columns of Tables 1- 4, where only Aeolus measurements with mid-points

within 25 km of the radars are included. For Rayleigh winds, there is no improvement in $ScMAD_z$ for the restricted data set, but the long along-track averaging distance of the Aeolus Rayleigh winds means that they still include contributions from up to 68 km away. For the Mie winds, with much shorter along-track averaging distance, there are improvements in $ScMAD_z$ with the restricted data set to 3.3 ms$^{-1}$ at MARA and 3.6 ms$^{-1}$ at ESRAD, well below the values for the other data subsets which are 4.2 - 6.4 ms$^{-1}$ or MARA, 3.9 - 4.1 ms$^{-1}$ for ESRAD. It seems likely that spatial variability is an important

contributor to $ScMAD_z$ , particularly at MARA. The geometry of the orbit passes at MARA means that 2 passes per week within 100 km are ascending, two descending. Only one (descending) pass per week comes within 25 km. The higher $ScMAD_z$ for ascending compared to descending passes at MARA could be explained by 50% of the descending passes being very close to the radar. Likewise, the much higher $ScMAD_z$ for winter compared to summer at MARA may simply reflect higher spatial variability of the winds in winter, particularly as the comparison is based primarily on measurements from the

lower troposphere. At ESRAD, three Aeolus orbits per week pass within 100 km, two descending (one to the East and one to the West) and one ascending (only the latter within 25 km). The only difference between the 25km dataset and the full ascending dataset is the along-orbit distance included in the averaging for the comparison. The small improvement of $ScMAD_z$ , from 4.0 to 3.6 ms$^{-1}$, with the restricted dataset suggests that spatial variability along the orbit path contributes a little at ESRAD. There is no difference no difference in $ScMAD_z$ between ascending and descending passes, suggesting

along-orbit and East-West spatial variability are about the same. The slightly higher $ScMAD_z$ for winter (4.1 ms$^{-1}$) compared to summer (3.9 ms$^{-1}$) may again be due to slightly higher spatial variability in winter. The higher values for $ScMAD_z$ for MARA compared to ESRAD (by 0.6 - 1.0 ms$^{-1}$) could be due to differences in local meteorology leading to differences in spatial variability. The higher $ScMAD_z$ for Rayleigh winds compared to Mie winds (by 1.0-1.4 ms$^{-1}$) is as could be expected because of different random errors in those wind estimates from Aeolus.

For the MARA and ESRAD data, Belova et al. (2021a) reported SD values for different subsets in the range 4-6 ms$^{-1}$ for Rayleigh winds, and mostly 3-5 ms$^{-1}$ for Mie winds. The present study shows $SD_z$ 5.5 - 6.8 ms$^{-1}$ for Rayleigh winds, 4.0 - 6.6 ms$^{-1}$ for Mie winds, which are somewhat higher. Figures 9 and 10 show how $bias_z$ and its confidence limits, $SD_z$ and $ScMAD_z$ vary over the two years of the present study. These show an overall increase in confidence limits for $bias_z$ for all cases, and in $SD_z$ and $ScMAD_z$ for Rayleigh winds. These are in line with the increase in estimated random errors for Aeolus

winds between June 2019 and June 2021 (2B11 baseline) shown by Lux et al. (2022) , which is due to degradation in power of the Aeolus lidar. There is no clear increase in $SD_z$ / ScMAD for the Mie wind comparison which could be due to the

bigger influence of spatial variability on those values. We note also that the precision of Mie winds should be less affected by laser-signal degradation as Mie winds are mainly retrieved form strong cloud scatter.

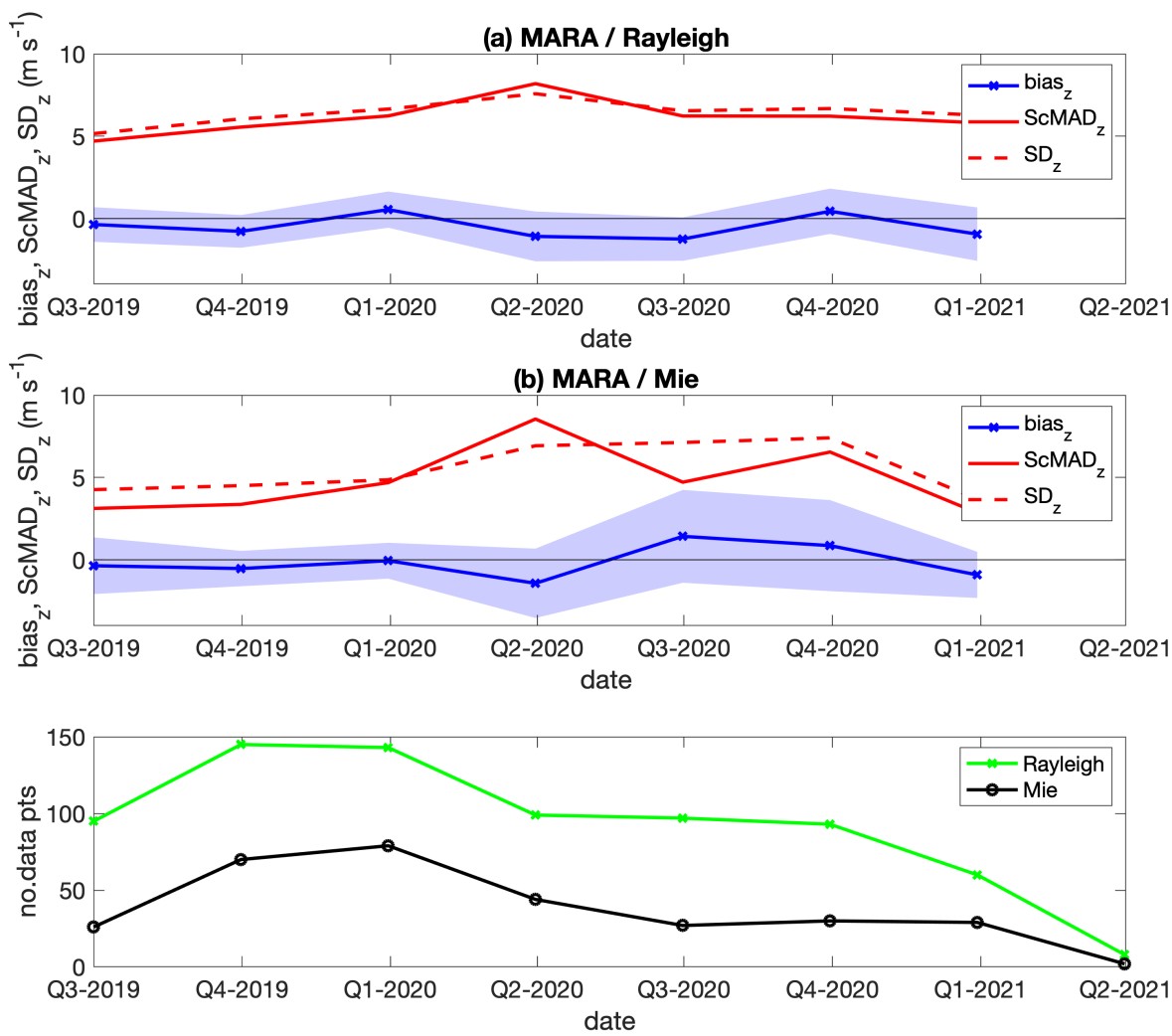

Fig 9. Variation over time of bias$_z$, ScMAD$_z$ and SD$_z$ for the Aeolus-MARA comparison. The shaded area around bias$_z$, indicates 95% confidence limits. Each quarter (all seasons, all orbits) over the 2-year study period was processed separately. Insufficient data is available for the last quarter. The bottom plot shows the number of data points that are included in the comparison.


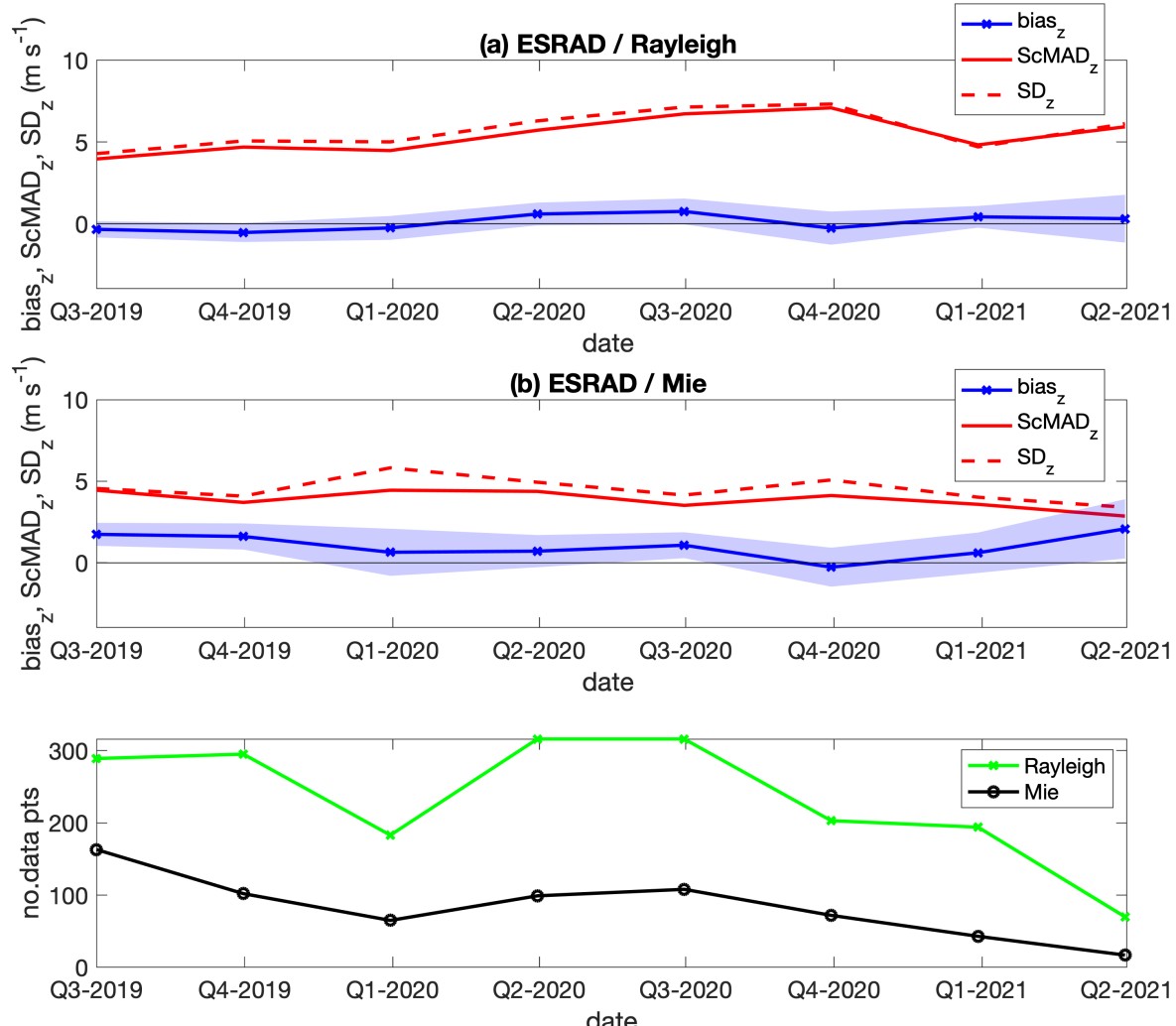


Fig 10. Variation over time of bias$_z$, ScMAD$_z$ and SD$_z$ for the Aeolus-ESRAD comparison. The shaded area around bias$_z$, indicates 95% confidence limits. Each quarter (all seasons, all orbits) over the 2-year study period was processed separately. The bottom plot shows the number of data points that are included in the comparison.


**Acknowledgements**

This work was supported by the Swedish National Space Agency (grant nos. 125/18 and 279/18). ESRAD operation and maintenance is provided by Esrange Space Center of Swedish Space Corporation. The team members at Maitri station for the Indian scientific expedition to Antarctica (ISEA) and the Antarctic logistics division at NCPOR (India) are acknowledged for providing necessary support for the operation of MARA.

## Author contributions

SK, EB and PV develop and maintain the software and data processing for ESRAD. SC and KS were responsible for operating the MARA radar and providing the data. SK, EB and PV developed the codes for the radar–Aeolus comparison and conducted the data analysis. SK , EB and PV prepared the paper with contributions from all co-authors.

## Financial support

This research has been supported by the Swedish National Space Agency (grant nos. 125/18 and 279/18).

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
