# Peer review of "Extended validation of Aeolus winds with wind-profiling radars in Antarctica and Arctic Sweden"

_EGUsphere, 2023_

## Author Comment (AC1)

Reply to reviewer #1

*Italics below indicate new text which is included in our revised manuscript.*

We thank the reviewer for their careful attention to the manuscript and for the suggestions for improvements.

General comments :

The reviewer asked for more comparison with other studies and more discussion of the contribution of radar measurement errors and other effects on the total random errors in the comparison between Aeolus winds. We have added more information and discussion as detailed below. The reviewer also asks for the evolution of the random errors over the 2-year period to be considered so we have added new figures dividing the data set into quarters and commented on this in the conclusions.

Specific comments:

1. Further references to Aeolus validation studies have been added

*Validation exercises soon after the start of the mission found that the quality of retrieved winds in part depended on the satellite's geolocation and on orbit orientation (see e.g. Guo et al, 2021; Lux et al., 2021). This could be traced back to unexpected instrumental effects, most prominently the influence of temperature on the performance of the primary telescope mirror of the instrument (Witschas et al., 2020; Lux et al., 2021; Weiler et al., 2021). The subsequent changes to the data-processing gave substantial improvement of the biases from more than 5m/s (Martin et al., 2021; Rennie and Isaksen, 2020) to less than 2 m/s (e.g. Iwai et al., 2021; Baars et al., 2022). However, Baars et al. (2022) noted that those improvements were partly masked by worsening instrument performance (e.g. decrease in laser output energy) that led to an increase of the random error. Nevertheless, Aeolus winds have been shown to make a positive contribution to global weather forecasting (Reitebuch et al,. 2020; Rennie et al., 2021; Weiler at al., 2021). A good number of validation comparisons of the corrected data processing after 2020 against a variety of other data sources have been reported, such as radiosondes (e.g. Martin et al. 2021; Rani et al., 2022; Chou et al., 2022), wind profiling radar (e.g. Guo et al, 2021; Kottayil et al., 2022; Chou et al., 2022), Doppler wind lidars (e.g. Chen et al., 2022, Witschas et al., 2022), numerical weather prediction models (e.g. Lux et al., 2022, Rani et al., 2022), and other satellites (Lukens et al., 2022). Overviews of recent validation comparisons were summarised by e.g. Wu et al (2022) and Ratynski et al. (2023), which mostly indicate possible biases less than 1 m/s and random errors 4-7 m/s for Rayleigh winds, 2-4 m/s for Mie winds. At the same time, the biases and random errors seem to vary more than might be expected between the different measurement techniques and locations used in the validations. Lux et al.(2022) have looked in detail at the non-random nature of differences between Aeolus winds and reference winds and suggest that the exact details of quality control applied in validation studies can significantly affect the results. They found that the bias and random error estimates can be affected by small num-*

*bers of outliers, particularly for Mie winds where large errors outside a Gaussian distribution (gross errors) can be caused by misinterpretation of noise as signal. This can lead to predominantly positively-biased gross errors.*

We have also updated the reference list accordingly.

2. In line 53 we add *(Moveable Atmospheric Radar for Antarctica)* after MARA , *(ESrange atmospheric RADar)* after ESRAD. We change the reference in line 55 to Belova et al. (2021a,b) and add :

*Each radar measures profiles of vertical and horizontal wind components in the vertical direction above the radar site. They switch automatically every 1-2 minutes between different modes with vertical resolution of 75 m, 150 m and 600 m (MARA)/ 900 m (ESRAD). The radars sample a cone of the atmosphere with a width of about 5° for ESRAD, 10° for MARA, so the horizontal diameter of the radar beams in the lowest 10 km of the atmosphere is less than 2 km at MARA, 1 km at ESRAD. Random errors (standard deviation of all 1 or 2-minute estimates in the 1-hour averages) are typically 2-3 ms−1 for both radars (Belova et al., 2021b). Comparison with radiosondes (Belova et al., 2021b). has shown no significant bias (<0.25 m s−1) for winds at MARA but systematic biases at ESRAD of 8% for zonal winds, 25% for meridional winds (ESRAD underestimates wind components). These are thought to be due to the geometry of the radar antenna field and a high level of local radio noise. The ESRAD wind estimates are corrected for these biases before being compared with Aeolus winds. For the comparison with Aeolus (as in Belova et al., 2021a), we use 1-hour averaged winds, also averaged over the height intervals corresponding to the Aeolus Rayleigh wind averages. We use only radar measurements where the 95% confidence limit of the 1-h mean is less than 2 m s−1 (this is calculated from the standard deviation and the number of the samples in the 1-h average, using students t-test).*

An extended discussion of the radar / spatial variability contribution to the random errors was added in section 6 (see point 7 further down).

We add

*Belova et al, 2021b : Belova, E., Voelger, P., Kirkwood, S., Hagelin, S., Lindskog, M., Körnich, H., Chatterjee, S., and Satheesan, K.: Validation of wind measurements of two mesosphere–stratosphere–troposphere radars in northern Sweden and in Antarctica, Atmos. Meas. Tech., 14, 2813–2825, https://doi.org/10.5194/amt-14-2813-2021, 2021b.*

and change Belova et al., 2021 to Belova et al., 2021a everywhere

3. See new paragraph added under 2.

4.  We add the total number of data points to the title of each panel in the figures so they become :

*Aeolus Rayleigh - MARA  N=907*
*Aeolus Mie - MARA  N=489*
*Aeolus Rayleigh - ESRAD  N=2093*
*Aeolus Mie - ESRAD N=1052*

and add to the Figure captions :
*N in the panel titles is the number of samples corresponding to n=100%.*

We also add the following to the description of Figure 1 in the text :

*The fraction of total comparison points left after applying both the EE and ModZ$_i$ rejection criteria (n$_z$) increases sharply for EE < 5 ms-1 and more slowly after that to just over 90% for Rayleigh (corresponding to ~1800 points) and to about 70% for Mie winds (~700 points).*

And to the description of Figure 2 in the text :

*The fraction of total comparison points left after applying both the EE and ModZ$_i$ rejection criteria (n$_z$) increases sharply for EE < 5 ms-1 and more slowly after that to just over 90% for Rayleigh (corresponding to ~1800 points) and to about 70% for Mie winds (~700 points).*

and add as a second last sentence in section 2

*This results in 80-90% of Rayleigh wind comparison points and about 60% of Mie wind points being available for analysis, sufficient numbers for further division according to summer/winter and ascending/descending orbits.*

5. The text in lines 110 and 138 is correct. The caption for Table 1 (MARA) will be corrected to

*(summer, 23 September - 22 March and winter 23 March - 22 September)*.

The caption for Table 3 (ESRAD) should read

*(winter, 23 September - 22 March and summer 23 March - 22 September)*

6. We can only guess at possible explanations for why the distribution is skewed - it's only for ESRAD Mie not Rayleigh, and only for ESRAD not MARA. It might, for example be to do with local meteorology . We add the following to the end of section 5.

*There is no obvious reason why the distribution is skewed, only for Mie winds, and only at ESRAD. One possibility might be local meteorology as the ESRAD area is often*

*covered by mountain-lee-wave clouds which might affect Mie (cloudy) measurements differently to Rayleigh (clear) ones.*

7. More discussion is now included in the conclusions.

To add to the discussion of the contribution of spatial differences, we have added a further column to each of Tables 1-4 which shows the results when we restrict the comparison to Aeolus measurements within 25 km of the radar sites, instead of 100 km. We add to the first paragraph of Section 3:

[revised manuscript text omitted]

As requested, we add a sentence to the first paragraph in section 3 to make clear that no correction is made for radar random errors - as we argue later these are small and the spatial variability contributes much more as discussed in the new paragraphs in Section 6.

*(Note that no correction is made in the Tables for the random uncertainties in radar measurements).*

8. We have divided the data into quarter-years and looked for changes over time (there are not enough points in the last quarter at MARA to include this). These are included as new Figures 9 an 10 (see below) and addressed in the final paragraph of the conclusions

*Figures  9 and 10 show how $bias_z$ and its confidence limits, $SD_z$ and $ScMAD_z$ vary over the two years of the present study.  These show an overall increase in confidence limits for $bias_z$ for all cases, and in $SD_z$ and $ScMAD_z$ for Rayleigh winds. These are in line with the increase in estimated random errors for Aeolus winds between June 2019 and June 2021 (2B11 baseline) shown by Lux et al. (2022) , which is due to degradation in power of the Aeolus lidar.  There is no clear increase in $SD_z$ / ScMAD for the Mie wind comparison which could be due to the bigger influence of spatial variability on those values.*

Technical corrections

We agree to all the technical corrections. For the missing units in Figures 3 and 5, we add this information to the caption to avoid cluttering the figures/tables.

Additional changes:

Revised Tables 1-4

Revised Figures 1,2,4,6,7,8

New figures 9,10

[Figure]

Fig 9. Variation over time of $bias_z$, $ScMAD_z$ and $SD_z$ for the Aeolus-MARA comparison. The shaded area around $bias_z$, indicates 95% confidence limits. Each quarter (all seasons, all orbits) over the 2-year study period was processed separately. Insufficient data is available for the last quarter.

[Figure]

Fig 10. Variation over time of bias$_z$, ScMAD$_z$ and SD$_z$ for the Aeolus-ESRAD comparison. The shaded area around bias$_z$, indicates 95% confidence limits. Each quarter (all seasons, all orbits) over the 2-year study period was processed separately.

---

## Author Comment (AC2)

Reply to reviewer #2

*Italics below indicate new text which is included in our revised manuscript.*

We thank reviewer #2 for their careful attention to the manuscript and for the suggestions for improvements.

General comments :

The reviewer asked for more comparison with other studies and more discussion of the contribution of radar measurement errors and other effects on the total random errors in the comparison between Aeolus winds. We have added more information and discussion as detailed below.

Specific comments:

1. We include more description of the radar characteristics and the contribution of various factors to the random errors. (see point 2 in the reply to reviewer 1)

To add to the discussion of the contribution of spatial differences, we have added a further column to each of Tables 1-4 which shows the results when we restrict the comparison to Aeolus measurements within 25 km of the radar sites, instead of 100 km. We add to the first paragraph of Section 3:

[revised manuscript text omitted]

As requested, we add a sentence to the first paragraph in section 3 to make clear that no correction is made for radar random errors - as we argue later these are small and the spatial variability contributes much more as discussed in the new paragraphs in Section 6.

*(Note that no correction is made in the Tables for the random uncertainties in radar measurements).*

2. We have added the number of available Aeolus measurements and the estimated radar error to the altitude profiles in Figures 4 and 6. We change the last paragraph of section 3 to :

*Figure 4 shows height resolved parameters for the Aeolus-MARA comparison. Fig. 4a and 4d shows that low heights between 1-5 km dominate the comparison even though Aeolus wind estimates are available throughout the troposphere (and higher in the*

*case of Rayleigh winds). This is due to the low sensitivity of the MARA radar in the up-*
*per troposhere and above. The uncertainty in radar winds is shown by the green line*
*in Fig. 4b and 4c. Each radar wind is estimated from a 1-h average of measurements*
*and the standard error of the mean (SEM) is used as an estimate of the uncertainty.*
*Since we include only averaged radar winds where the 95% confidence interval is <*
*$2ms^{-1}$ (this is twice the SEM when the number of data points in the average is large) ,*
*SEM is low, below 1 $ms^{-1}$ and increases only slightly with height. (The $SEM_{MARA}$ profile is*
*essentially the same for the ascending and descending passes as for all data so, for*
*clarity, it is not included in the plot.) In Figs. 4e and 4f we can see that the negative*
*bias for Rayleigh descending winds, seen in Tables 1 and 2, is seen at almost all*
*heights, although the uncertainties in the bias become very large above 6 km height.*
*It is partly balanced by a positive bias (marginally significant) for the ascending*
*passes so that, for all data together (Figs. 4b and 4e), the mean bias becomes closer*
*to zero. For the Mie winds, with notably more restricted height coverage, there is no*
*significant bias at any height.*

and the last paragraph of Section 4 to :

*Figure 6 provides height-resolved profiles of parameters for the Aeolus-ESRAD com-*
*parison. As can be seen in Figs 5a and 5d, in contrast to MARA, the more powerful ES-*
*RAD radar provides useful coverage in the upper troposphere  as well as the lower tro-*
*posphere. There are fewer joint ESRAD-Aeolus observations than Aeolus alone al-*
*though, between 2 and 5 km height almost all Aeolus measurements have corre-*
*sponding radar ones, about half higher up in the troposphere. The green line in Figs*
*5b and 5e shows the mean SEM for the ESRAD wind averages, reaching 1 $ms^{-1}$ at in*
*the upper troposphere, lower at lower heights. (The $SEM_{ESRAD}$ profile is essentialy the*
*same for the ascending and descending passes as for all data so, for clarity, it is not*
*included in the plot.)  Considering the bias profiles in Figs 4b,c,e,f , above 6 km height,*
*the bias uncertainties are notably lower than at MARA - this is a result of a much*
*larger number of comparison points thanks to the higher power of the ESRAD radar.*
*For Rayleigh winds there is no significant bias at any height, for Mie winds the ~1 m/s*
*positive bias identified in Table 4 is clearly seen at all heights. From Fig. 2 it is clear*
*that a positive bias appears whatever the EE threshold.*

3. Reference to Belova et al., 2021a added and HLOS capitalised everywhere

4. 'for all available data' added to the figure captions, total number given in panel title and EE limit used in further statistics marked.

5. Reference to Belova et al., 2021a added

6. We have made corresponding plots to Figs 1 and 2 for each data subset separately and these show essentially the same behaviour as Figs. 1 and 2, except for the levels of the bias and SD/ScMad values. These are too many to include in the paper but we add the following text in the last paragraph of Section 2.

*We have made similar plots for all of the data subsets which we analyse below and found no reason to choose different thresholds for the different subsets. In all cases, ScMAD is close to $ScMAD_z$ and their values are constant or changing very slowly between EE values 1 ms-1 above or below the thresholds. Similarly, bias and $bias_z$ are close together and insensitive to the EE values around the chosen thresholds, although both the $bias_z$ and $ScMAD_z$ values can lie at different levels in the different subsets, as shown in Tables 1 and 2 and discussed in the next section.*

7. We have added more discussion of the random errors in the 3rd paragraph of section 6 (see point 1 above)

8. We have added a sentence on seasonal effects, citing Weiler et al.

9. In line 114 we say "For the Rayleigh winds, Table 1 shows that there are no significant differences between summer and winter." The bias 95% confidence limits for summer are [ -0.6 0.6 ] and for winter [-1.6 0.1]. These have a large overlap which we interpret as no significant difference.

Technical corrections:

1. Caption corrected and units added to caption

2. EE thresholds added to captions

3. We have added the text, units and a row showing the range of wind speeds to Tables 1-4.

4. The scale has been expanded

5. X-axis has been relabelled

Additional changes:

Revised Tables 1-4

Revised Figures 1,2,4,6,7,8

New figures 9,10

[Figure]

Fig 9. Variation over time of $bias_z$, $ScMAD_z$ and $SD_z$ for the Aeolus-MARA comparison. The shaded area around $bias_z$, indicates 95% confidence limits. Each quarter (all seasons, all orbits) over the 2-year study period was processed separately. Insufficient data is available for the last quarter.

[Figure]

Fig 10. Variation over time of bias$_z$, ScMAD$_z$ and SD$_z$ for the Aeolus-ESRAD comparison. The shaded area around bias$_z$, indicates 95% confidence limits. Each quarter (all seasons, all orbits) over the 2-year study period was processed separately.

---

## Author Response (AR2)

**Reply to editor**

*Italics below indicate new text which is included in our revised manuscript.*

The following changes were made:

- Figs. 9 and 10 were modified to include plots with the numbers of data points. Figure captions were modified by adding a sentence:
  *The bottom plot shows the number of data points that are included in the comparison.*
- Figs. 7 and 8 were modified by changing the labels on the horizontal axes. They read now:
  *Rayleigh wind error with respect to MARA*
  and
  *Mie wind error with respect to ESRAD.*

Additional changes were made based on comments by both reviewers.

Reviewer #1:

1. When discussing the fact that the Mie random error did not change much over time as opposed to the Rayleigh random error (l. 443 f.), the authors should add that the Mie wind precision was less affected by the signal degradation of Aeolus, as the Mie winds are mainly retrieved from strong cloud backscatter.

We add a sentence at the end of section 6 (l 452):
*We note also that the precision of Mie winds should be less affected by laser-signal degradation as Mie winds are mainly retrieved form strong cloud scatter.*

2. I also noticed that the authors added a line to Tables 1 through 4 providing the median Aeolus HLOS wind speed in each data subset as well as the lower and upper quartiles of the distribution. However, they do not refer to these statistical parameters in the text. I suggest to discuss the variation of the added parameters among the different data subsets or remove the corresponding lines from the tables.

This information was added to the tables at the request of the other reviewer.

Reviewer #2:

Specific comments:

1. Please make a general statement or add to the fig. captions, to clarify if 100 km / 25 km collocation data is the source for the respective plots.

We added two sentences to adress this point:
*(Note that results restricted to Aeolus measurements only within 25 km from the radar are shown only in the Tables. All of the figures include points up to 100 km from the radar)*

2. L 364: … Could you please check for vertical wind speeds at the ESRAD site and add a sentence on a possible contribution or explanation for the skewed distribution.

The contribution of vertical components was checked and found to be negligible - this is already stated on line 97: " Radar 'HLOS' winds are calculated from the radar vector winds (ignoring the vertical component, which is found to be negligible in the 1 h averages) ". We added additional explanation after l 364:
*In general, vertical winds of up to 2 m/s can be found in the troposphere in mountain lee waves at ESRAD (Kirkwood et al., 2010). However, the horizontal wavelengths of the lee waves are only a few 10s of km and would be averaged along the Aeolus track. In the comparison data set here, 99% of the data points have vertical winds within +0.4/-0.4 m/s at ESRAD and there is no correlation between vertical wind and the difference between ESRAD and Aeolus HLOS winds. So vertical winds cannot explain the skewed distribution. Preferential locations for cloud formation within the wave wind field could affect Mie winds differently from Rayleigh winds. Extensive case studies would be needed to test this possibility.*

The reference Kirkwood et al. (2010) was added to the list of references at the end of the manuscript.

*3.* L444: For Q2-2020 the ScMAD stands out in Fig 9, and is unexpectedly higher than the SD. Please check for consistency in the data analysis and elaborate on this. Please add the respective evolution of the number of data points to Fig. 9 and 10 or a general comment (# > x, e.g. 50), which also might be part of an explanation or reason to flag this data as an outlier or even statistically non-significant.

The number of data points is added to the figures 9 and 10. Although not very high for the points in question ( 99 / 44 for  Rayleigh/Mie for Q2-2020 in Fig. 9) these are not low enough to mean the data should be rejected.

4. Please consider adding a sentence or two on the outlook, e.g. if you would consider re-validating reprocessed data, for which your long time-series, high latitude radar validation would be a very valuable contribution…

Funding and personnel constraints are unlikely to allow us to make further analyses at this time, and future operation of both radars is very uncertain so we prefer not to comment on the outlook.

Specific comments:

a) L20: Please consider adding "2B11 data is considered…" or otherwise refer to the major used baseline already in the abstract.

We added a paranthesis to indicate the baseline we considered.

b) – d) We corrected the manuscript accordingly.

e) Screen for units to not be disconnected from the values during line breaks.

We tried to correct all instances we found. However, this seems to be an artifact of the layout that is provided by the word processor software and that is not always correctable.